# Sam68/KHDRBS1-dependent NF-κB activation confers radioprotection to the colon epithelium in γ-irradiated mice

Kai Fu[1], Xin Sun[1], Eric M Wier[1], Andrea Hodgson[1], Ryan P Hobbs[1], Fengyi Wan[1,2,3]*

[1]Department of Biochemistry and Molecular Biology, Bloomberg School of Public Health, Johns Hopkins University, Baltimore, United States; [2]Department of Oncology, School of Medicine, Johns Hopkins University, Baltimore, United States; [3]The Sidney Kimmel Comprehensive Cancer Center, Johns Hopkins University, Baltimore, United States

**Abstract** Previously we reported that Src-associated-substrate-during-mitosis-of-68kDa (Sam68/KHDRBS1) is pivotal for DNA damage-stimulated NF-κB transactivation of anti-apoptotic genes (Fu et al., 2016). Here we show that Sam68 is critical for genotoxic stress-induced NF-κB activation in the γ-irradiated colon and animal and that Sam68-dependent NF-κB activation provides radioprotection to colon epithelium in vivo. Sam68 deletion diminishes γ-irradiation-triggered PAR synthesis and NF-κB activation in colon epithelial cells (CECs), thus hampering the expression of anti-apoptotic molecules in situ and facilitating CECs to undergo apoptosis in mice post whole-body γ-irradiation (WBIR). Sam68 knockout mice suffer more severe damage in the colon and succumb more rapidly from acute radiotoxicity than the control mice following WBIR. Our results underscore the critical role of Sam68 in orchestrating genotoxic stress-initiated NF-κB activation signaling in the colon tissue and whole animal and reveal the pathophysiological relevance of Sam68-dependent NF-κB activation in colonic cell survival and recovery from extrinsic DNA damage.

*For correspondence: fwan1@jhu.edu

**Competing interests:** The authors declare that no competing interests exist.

## Introduction

Nuclear factor kappa B (NF-κB) plays a crucial function in a variety of human disorders, in particular inflammatory diseases and cancers (*Hayden and Ghosh, 2008*; *Scheidereit, 2006*; *Sun et al., 2013*; *Vallabhapurapu and Karin, 2009*; *Wan and Lenardo, 2010*; *Wu and Miyamoto, 2007*). Accumulating evidence highlights an important role of NF-κB signaling pathway in cellular responses to various genotoxic stresses and DNA damage-stimulated NF-κB signaling cascade in the nucleus that leads to NF-κB activation has been recently revealed (*McCool and Miyamoto, 2012*; *Miyamoto, 2011*). In particular, ataxia telangiectasia mutated (ATM), inhibitor of NF-κB kinase gamma subunit (IKKγ), protein inhibitor of activated STATy (PIASy), and poly (ADP-ribose) polymerase 1 (PARP1) were reported to be indispensible for genoxic stress-induced NF-κB activation (*Huang et al., 2003*; *Li et al., 2001*; *Mabb et al., 2006*; *Piret et al., 1999*; *Stilmann et al., 2009*). Moreover, we recently revealed that Sam68/KHDRBS1 (Src-associated substrate during mitosis of 68 kDa/KH domain containing, RNA binding, signal transduction associated 1, encoded by *KHDRBS1* gene), a versatile single-strand nucleic acid binding protein (*Lukong and Richard, 2003*; *Richard, 2010*), is an important molecule in orchestrating genotoxic stress-initiated NF-κB signaling in the nucleus (*Fu et al., 2016*). Specifically, Sam68 is essential for DNA damage-triggered PARP1 activation and the subsequent polymers of ADP-ribose (PAR) synthesis (*Fu et al., 2016*). Sam68 deletion dampens the PAR-

dependent NF-κB signaling and transcription of an array of anti-apoptotic genes, thus sensitizing Sam68-deficient mouse embryonic fibroblasts (MEFs) and colon epithelial cells (CECs) in culture to genotoxicity caused by DNA-damaging agents (*Fu et al., 2016*). The levels of Sam68, PAR, NF-κB activation, and anti-apoptotic molecules B-cell lymphoma-extra large (Bcl-XL) and X-linked inhibitor of apoptosis protein (XIAP) are elevated and positively correlated in colon tumors compared to adjacent normal tissue derived from either the tumor-laden $Apc^{min716/+}$ mice or human colon cancer patients. Moreover, downregulation of Sam68 substantially sensitizes human colon cancer cells to spontaneous and genotoxic stress-induced cell death and retards colon tumor burden in $Apc^{min716/+}$ mice (*Fu et al., 2016*). These findings suggest that upregulated Sam68 is crucial in orchestrating DNA damage-initiated NF-κB activation signaling in cultured cells and conferring the PAR-dependent NF-κB activation to respond to the intrinsic DNA damage frequently occurred in cancerous cells. However, the in vivo impact of physiological Sam68 levels on extrinsic genotoxic stress-induced NF-κB signaling and activation in normal cells at the organ and even the whole animal levels has not been fully understood.

Radiotherapy and chemotherapy are extensively used in current-day cancer treatments. It has been recognized that γ-irradiation induced DNA damage triggers rapidly-proliferating tumor cells to undergo apoptosis; whereas non- and slowly-dividing cells rarely die of γ-irradiation. Of note, CECs in colon crypts are among the most rapidly dividing cells in the body, which makes them susceptible to γ-irradiation-induced cell death. Indeed, colon tissue injury remains one of the major adverse effects of radiotherapy, when γ-irradiation is employed to treat colon cancer and other intra-abdominal cancers (*Egan et al., 2004*). Although greater antitumor effects could be produced by higher doses of γ-irradiation, the tolerance of patients to the acute side effects caused by γ-irradiation to the colon limits the administered dose (*Egan et al., 2004*). Given the essential role of DNA damage-induced NF-κB transactivation of anti-apoptotic genes in cell fate determination post genotoxic stresses, it will be extremely important to understand genotoxic stress-induced NF-κB activation signaling pathway not only in cancerous cells but also in normal cells under pathophysiological conditions. Besides the recently revealed critical function of Sam68-dependent NF-κB activation to overcome intrinsic DNA damage for the development and survival of colon cancer (*Fu et al., 2016*), whether Sam68-dependent NF-κB signaling is crucial in normal colon epithelium in response to extrinsic γ-irradiation remains elusive. We therefore examined the hypothesis that Sam68-dependent NF-κB activation offers an anti-apoptotic response in the γ-irradiated colon epithelium in vivo hence providing radioprotection to the organ and the animal following whole-body γ-irradiation (WBIR).

## Results

### Sam68 confers genotoxic stress-induced NF-κB signaling in the γ-irradiated colon

To assess the in vivo impact of Sam68 on genotoxic stress-induced NF-κB signaling and transactivation, $Khdrbs1^{+/-}$ (Sam68 heterozygote) and $Khdrbs1^{-/-}$ (Sam68 knockout) mice were subjected to a sublethal dose of WBIR and we examined the γ-irradiation-initiated NF-κB activation signaling cascade in the derived colons at defined times post WBIR (*Figure 1A*). As expected, vigorous PAR production, as illustrated by immunofluorescence staining on colon tissue sections, occurred in $Khdrbs1^{+/-}$ colon 20 min post WBIR; whereas such an acute response was almost abolished in the colon from $Khdrbs1^{-/-}$ mice (*Figure 1B*). In support, immunoblot analyses showed that robust PAR chair formation in whole cell lysates of CECs from $Khdrbs1^{+/-}$ mice at 20 min post WBIR, which was markedly tempered in $Khdrbs1^{-/-}$ mice post WBIR (*Figure 1C*). These results suggest that Sam68 is crucial for facilitating genotoxic stress-induced PAR production in the γ-irradiated colon tissue in vivo. Consistently, WBIR-induced p65 phosphorylation, one of the biochemical hallmarks of NF-κB activation, was profound in the CECs from γ-irradiated $Khdrbs1^{+/-}$ mice, but was substantially tempered in $Khdrbs1^{-/-}$ animals (*Figure 1D*). Moreover, WBIR triggered nuclear translocation of p65, as assayed by immunohistostaining and subcellular fractionation, in $Khdrbs1^{+/-}$ CECs; whereas p65 nuclear accumulation was greatly attenuated in the γ-irradiated $Khdrbs1^{-/-}$ cells (*Figure 1E–F*). Of note, the levels of PAR, total p65, and nuclear accumulated p65 were comparable in CECs derived from mock-irradiated $Khdrbs1^{+/-}$ and $Khdrbs1^{-/-}$ animals (*Figure 1B–F*), suggesting that Sam68 deletion does not affect the physiological PAR synthesis and NF-κB signaling in the colon without

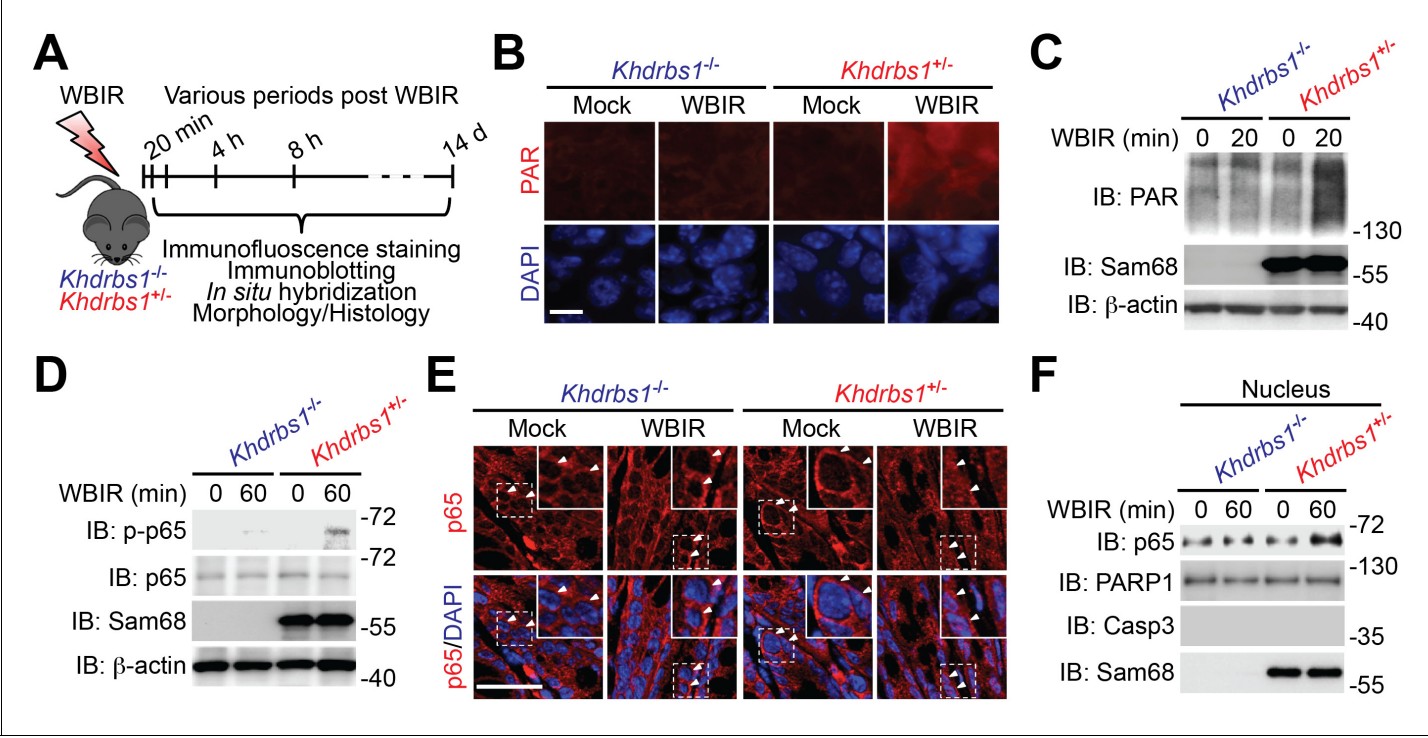

**Figure 1.** Sam68 deletion diminishes genotoxic stress-induced NF-κB signaling in the γ-irradiated colon. (**A**) A schematic of the experimental timeline for the impact of Sam68 deletion on DNA damage-induced NF-κB signaling pathway in γ-irradiated mice. *Khdrbs1*$^{+/-}$ and *Khdrbs1*$^{-/-}$ mice subjected to a sublethal dose (6.5 Gy) of whole body γ-irradiation (WBIR) or mock irradiation were euthanized at the indicated periods post WBIR, followed by the analyses as indicated. (**B**) Immunofluorescence micrographs of PAR in colon tissue collected from *Khdrbs1*$^{+/-}$ and *Khdrbs1*$^{-/-}$ mice at 20 min following WBIR or mock irradiation, with nuclei counterstained by DAPI. Scale bar, 25 μm. (**C and D**) Colon epithelial cells (CECs) were isolated from *Khdrbs1*$^{+/-}$ and *Khdrbs1*$^{-/-}$ mice at the indicated periods post WBIR, and whole cell lysates were derived and immunoblotted (IB) for indicated proteins, with β-actin as a loading control. p-p65, phosphorylated p65. (**E**) Immunofluorescence micrographs of p65 in colon tissue collected from *Khdrbs1*$^{+/-}$ and *Khdrbs1*$^{-/-}$ mice at 60 min post WBIR. Scale bar, 50 μm. (**F**) CECs were collected from *Khdrbs1*$^{+/-}$ and *Khdrbs1*$^{-/-}$ mice as treated in (**E**) and nuclear fractions were derived and IB for indicated proteins. Caspase-3 (Casp3) and PARP1 served as loading controls and cytosolic and nuclear markers, respectively.

any stimulation. In contrast, Sam68 deletion almost abolished genotoxic stress-triggered PAR formation and the signaling events that lead to NF-κB activation in the colon following WBIR (*Figure 1B–F*). Consistent with our recent report that Sam68 plays a key role in DNA damage-initiated PAR synthesis and the PAR-dependent NF-κB signaling in the isolated and in vitro cultured CECs (*Fu et al., 2016*), these results further support the crucial function of Sam68 in genotoxic stress-triggered PAR production and signaling to NF-κB activation in the γ-irradiated colon from whole animals in vivo.

## Sam68 is critical for anti-apoptotic gene transcription in the γ-irradiated colon

It has been well established that NF-κB mediates the gene transcription of a panel of anti-apoptotic molecules in cells following genotoxic stress (*Fu et al., 2016*; *Kim et al., 2005*; *Stilmann et al., 2009*). We therefore assessed the impact of Sam68 on γ-irradiation-induced expression of NF-κB target gene *Bcl2l1*, which encodes B-cell lymphoma-extra large (Bcl-XL). As illustrated by digoxigenin-labeled messenger RNA (mRNA) in situ hybridization, *Bcl2l1* mRNA levels were elevated in colon tissue sections derived from whole-body γ-irradiated *Khdrbs1*$^{+/-}$ mice (*Figure 2A*). In contrast, WBIR-induced transcription of *Bcl2l1* was substantially tempered in the γ-irradiated *Khdrbs1*$^{-/-}$ colon tissue (*Figure 2A*). In line with the nearly abolished γ-irradiation-initiated NF-κB signaling cascade in the colon of *Khdrbs1*$^{-/-}$ mice post WBIR (*Figure 1B–F*), these results demonstrate that Sam68 deletion suppresses the inducible transcription of NF-κB target genes in the colon in situ following WBIR. Mirroring the robust transcription of *Bcl2l1* triggered by γ-irradiation (*Figure 2A*), Bcl-XL

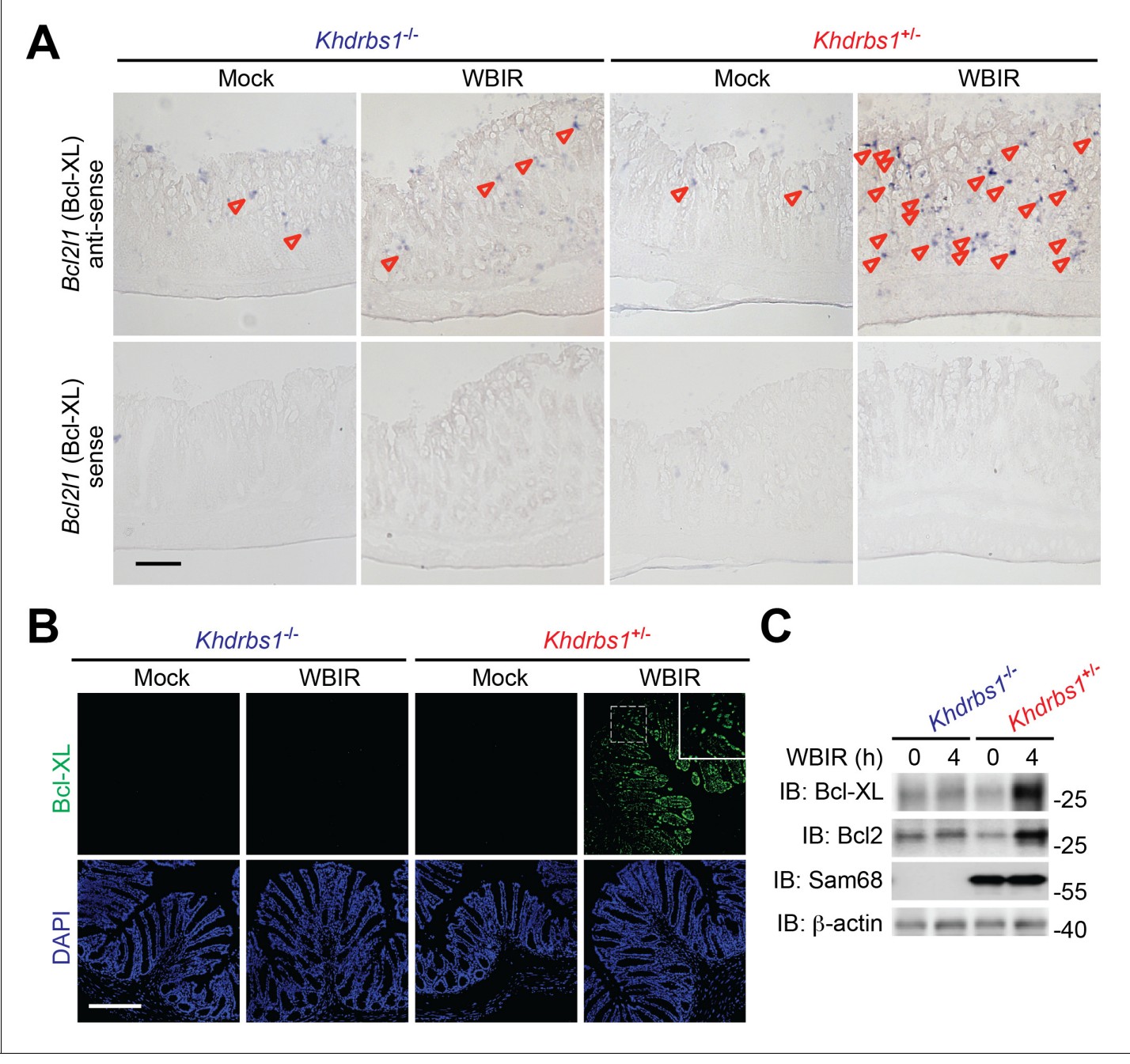

**Figure 2.** Sam68 is pivotal for NF-κB-mediated anti-apoptotic gene expression in the γ-irradiated colon. (**A**) Colon tissue sections derived from *Khdrbs1$^{+/-}$* and *Khdrbs1$^{-/-}$* mice at 4 hr post whole body γ-irradiation (WBIR) or mock irradiation were stained by in situ hybridization with in vitro synthesized anti-sense probe targeting *Bcl2l1* mRNA (purple dots as indicated by triangles), with *Bcl2l1* mRNA sense probe as a negative control. Scale bar, 100 μm. (**B**) Immunofluorescence micrographs of Bcl-XL (encoded by *Bcl2l1*) in colon tissue collected from mice treated as in (**A**), with nuclei counterstained by DAPI. Scale bar, 200 μm. (**C**) Colon epithelial cells were isolated from mice treated as in (**A**) and whole cell lysates were derived and immunoblotted (IB) for indicated proteins, with β-actin as a loading control.

protein levels were also induced in the colon tissue derived from *Khdrbs1$^{+/-}$* mice at 4 hr post WBIR (*Figure 2B*). In striking contrast, WBIR-induced Bcl-XL upregulation was diminished in the colon derived from the whole-body γ-irradiated *Khdrbs1$^{-/-}$* animals (*Figure 2B*). Moreover, these results were further supported by immunoblot of Bcl-XL and another anti-apoptotic protein B-cell

lymphoma 2 (Bcl2), encoded by the NF-κB target gene *Bcl2*, in the CEC lysates isolated from the whole-body γ-irradiated mice (*Figure 2C*). Hence Sam68 is essential for genotoxic stress-induced and NF-κB-mediated expression of anti-apoptotic genes in the γ-irradiated colon epithelium.

## Sam68-deleted colon epithelial cells are more sensitive to whole-body γ-irradiation

The balance between severe DNA damage-triggered programmed cell death and genotoxic stress-induced NF-κB-mediated anti-apoptotic transcription is pivotal for cell fate determination in cellular responses to DNA-damaging agents (*Fu et al., 2016*; *Kim et al., 2005*; *Stilmann et al., 2009*). We barely detected the cleavage of Caspase-3, one well-established biochemical hallmark for apoptosis, in the colon tissue derived from *Khdrbs1*$^{+/-}$ mice at 8 hr post WBIR (*Figure 3A*), as supported by the evidence that WBIR triggered profound NF-κB activation signaling (*Figure 1*) and expression of anti-apoptotic molecules Bcl-XL and Bcl2 (*Figure 2*) in Sam68-sufficient CECs. In contrast, Caspase-3 cleavage was substantially augmented in the γ-irradiated *Khdrbs1*$^{-/-}$ colon (*Figure 3A*), which correlates with the diminished NF-κB signaling in the nucleus (*Figure 1*) and inefficient anti-apoptotic gene expression (*Figure 2*) in the absence of Sam68. Moreover, immunoblot analyses of the CEC lysates further ascertained that the elevation in cleaved Caspase-3 and cleaved PARP1, another known biochemical hallmark for apoptosis, occurred in the whole-body γ-irradiated *Khdrbs1*$^{-/-}$ mice, but not *Khdrbs1*$^{+/-}$ controls (*Figure 3B*). Such an inverse correlation between NF-κB-mediated anti-apoptotic gene expression and DNA damage-triggered apoptosis underscores the crucial function of Sam68 in genotoxic stress-induced NF-κB signaling and transactivation in the γ-irradiated colon epithelium. Consistently, far more apoptotic cells, as assayed by terminal deoxynucleotidyl transferase dUTP nick end labeling (TUNEL), were observed in the colon tissue sections in situ from *Khdrbs1*$^{-/-}$ mice post WBIR, compared to those from *Khdrbs1*$^{+/-}$ controls (*Figure 3C–D*). The amount of apoptotic cells on the colon tissue sections from the mock-γ-irradiated animals was comparable, regardless of Sam68 presence (*Figure 3C–D*). These results thus demonstrate that Sam68 deletion expedites CECs to undertake apoptosis in vivo, in parallel to the substantially dampened NF-κB signaling and anti-apoptotic gene expression caused by genotoxic stress, in the mice subjected to WBIR.

## Sam68 is crucial for the NF-κB-mediated radioprotection in the colon of γ-irradiated animals

Previous studies reveal that the intestine and the colon are hypersensitive to radiotoxicity (*Barlow et al., 1996*; *de Murcia et al., 1997*; *Gannon et al., 2012*) and that NF-κB signaling pathway executes an important protective function in the γ-irradiated colon (*Egan et al., 2004*). To assess the impact of Sam68 on the radiodamage to the colon tissue, we examined the morphology of the colon from mice, relative to mock-treated controls, by gross dissection and histological staining. Indeed, the colonic morphology, length, and structure of mock-irradiated *Khdrbs1*$^{+/-}$ and *Khdrbs1*$^{-/-}$ mice were indistinguishable, suggesting that Sam68 is dispensable for mouse colon development (*Figure 4A–E*). Fourteen days post WBIR, the colons in *Khdrbs1*$^{+/-}$ mice were comparable to those from mock-irradiated animals in morphology and length (*Figure 4A–C*). In contrast, the γ-irradiated *Khdrbs1*$^{-/-}$ mice, compared to *Khdrbs1*$^{+/-}$ controls, suffered more severe and widespread damage to the colon, with substantially shortened colon lengths (*Figure 4A–C*). Moreover, histological analyses showed more severe crypt shrinkage, more goblet cell depletion, and less crypt survival in the colon derived from *Khdrbs1*$^{-/-}$ mice than in those from *Khdrbs1*$^{+/-}$ controls following WBIR (*Figure 4D–E*). Consequently, far fewer *Khdrbs1*$^{-/-}$ mice survived the sublethal dose of WBIR, compared to *Khdrbs1*$^{+/-}$ controls (*Figure 4F*), demonstrating that Sam68 deletion promotes the mice to be hypersensitive to radiotoxicity. Consistent with the reported crucial role of NF-κB signaling for providing radioprotection to the colon epithelium (*Egan et al., 2004*), our results emphasize that Sam68 executes a key function in genotoxic stress-induced NF-κB signaling and transactivation of a panel of anti-apoptotic genes, thus conferring radioprotection to the colon in the whole-body γ-irradiated mice.

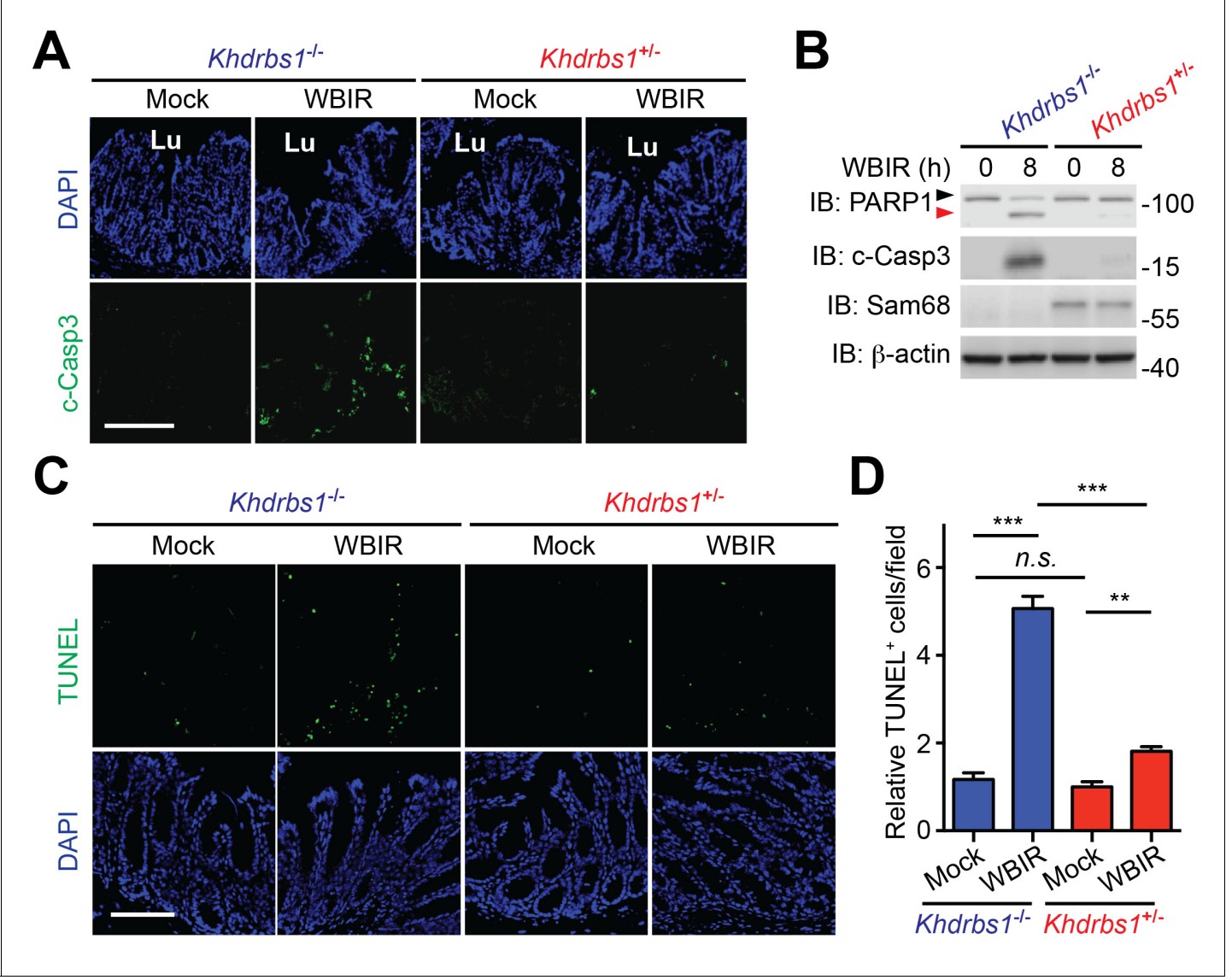

**Figure 3.** Sam68 deletion sensitizes colon epithelial cells to undergo apoptosis in the γ-irradiated mice. (A) Immunofluorescence micrographs of cleaved Caspase-3 (c-Casp3) in colon tissue collected from *Khdrbs1*$^{+/-}$ and *Khdrbs1*$^{-/-}$ mice at 8 hr post whole body γ-irradiation (WBIR) or mock irradiation, with nuclei counterstained by DAPI. Lu, lumen; Scale bar, 200 μm. (B) Colon epithelial cells were isolated from mice treated as in (A) and whole cell lysates were derived and immunoblotted (IB) for indicated proteins, with β-actin as a loading control. The full-length and cleaved PARP1 are indicated by a black triangle and a red triangle, respectively. (C) Micrographs of TUNEL staining in colon tissue collected from mice treated as in (A), with nuclei counterstained by DAPI. Scale bar, 100 μm. (D) Relative cells with TUNEL staining from four random fields, as in (C), were quantified. Data are representative of at least two independent experiments. Results in (D) are expressed as mean and s.e.m. n.s., non-significant difference and **p<0.01, ***p<0.001 by Student's *t* tests.

## Discussion

Herein, we report that Sam68 is critical for γ-irradiation-initiated NF-κB signaling and anti-apoptotic transcription in the colon in vivo and that Sam68-dependent NF-κB activation executes a protective function to the colon epithelium in the whole-body γ-irradiated animals. Sam68 deletion substantially dampens the γ-irradiation-initiated signaling cascade essential for NF-κB activation, which includes PAR synthesis, p65 phosphorylation, and p65 nuclear translocation, in the colon derived from mice at various time periods post WBIR. As a consequence, γ-irradiation-induced expression of NF-κB tar-get genes, in particular *Bcl2l1* encoding the anti-apoptotic protein Bcl-XL, is remarkably tempered in

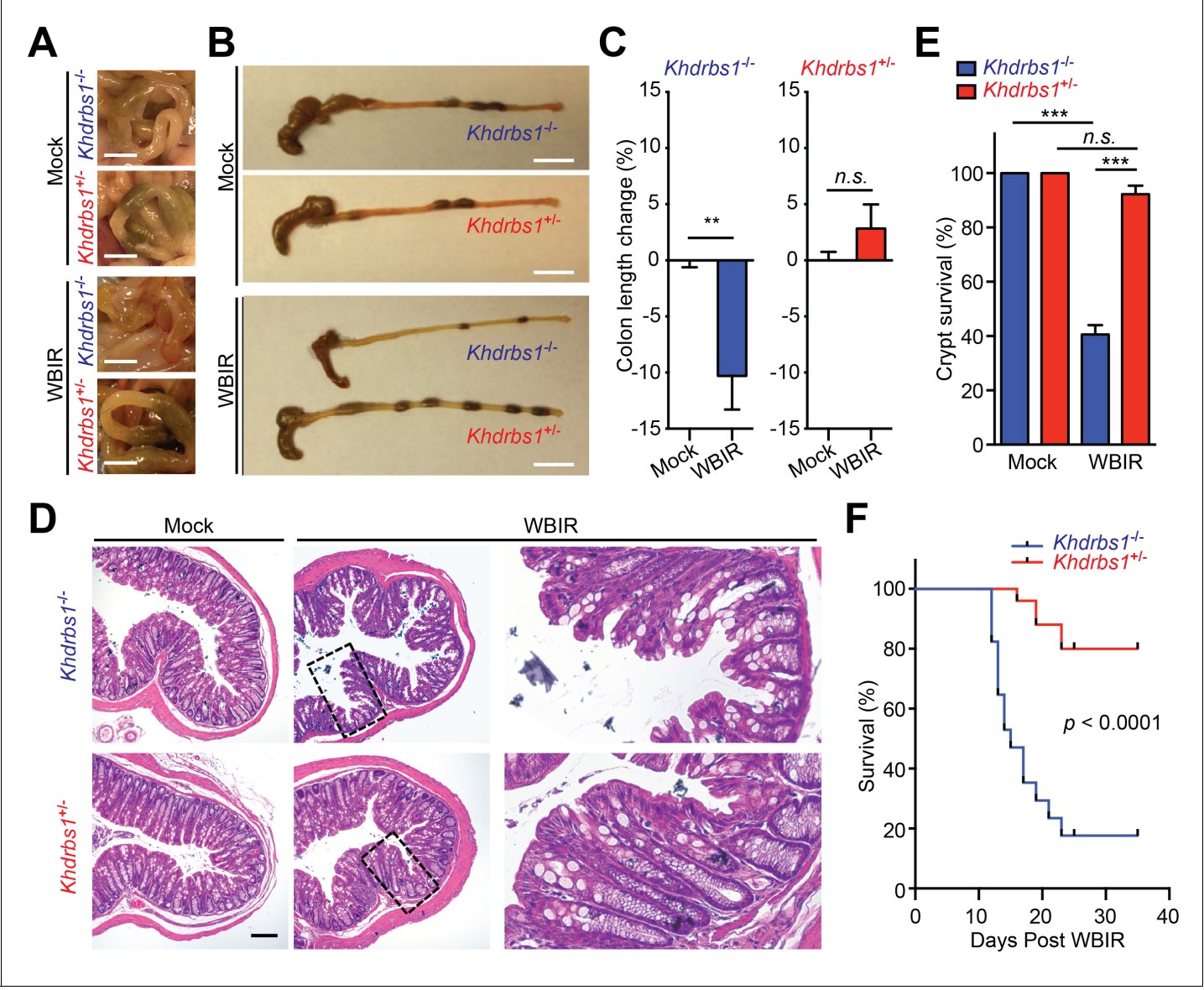

**Figure 4.** Sam68 is essential for the NF-κB-mediated radioprotection in vivo. (A and B) Representative photographs of colons in (A) and collected from (B) *Khdrbs1*[+/−] and *Khdrbs1*[−/−] mice at 14 days post whole body γ-irradiation (WBIR) or mock irradiation. Scale bars, 5 mm (A) and 1 cm (B), respectively. (C) The length changes in the colons derived from *Khdrbs1*[+/−] (n = 3) and *Khdrbs1*[−/−] (n = 3) mice at 14 days post WBIR or mock-irradiation, normalized to mock-irradiated controls. (D) Hematoxylin and eosin staining of colon tissue sections collected from mice treated as in (A). Scale bar, 100 μm. (E) Percentage of crypt survival in (D) was quantified. (F) Kaplan-Meier analysis of the survival rate in *Khdrbs1*[+/−] (n = 25) and *Khdrbs1*[−/−] (n = 17) mice following WBIR. p<0.0001 by Gehan-Breslow-Wilcoxon test. Results in (C and E) are expressed as mean and s.e.m. n.s., non-significant difference and **p<0.01, ***p<0.001 by Student's *t* tests.

the colon epithelium from *Khdrbs1*[−/−] mice, compared to *Khdrbs1*[+/−] controls. These results are consistent with our prior report that Sam68 deletion diminishes the genotoxic stress-induced NF-κB signaling and NF-κB-mediated anti-apoptotic gene expression in the cultured MEFs and CECs in vitro (*Fu et al., 2016*). Moreover, WBIR fosters *Khdrbs1*[−/−] CECs to undertake apoptosis in situ in the colon from *Khdrbs1*[−/−] mice, but not *Khdrbs1*[+/−] controls, which also mirrors our prior report that *Khdrbs1*[−/−] CECs are hypersensitive to γ-irradiation and other genotoxic stresses in culture (*Fu et al., 2016*). Our results generated from whole-body γ-irradiated animals, along with our previous reports in the cultured cells, further support the physiological relevance of Sam68 in

orchestrating genotoxic stress-initiated NF-κB activation signaling in the colon epithelium in response to genotoxic stresses. As elucidated previously (*Fu et al., 2016*), knockdown/knockout of Sam68 substantially sensitizes human colon cancer cells to undergo spontaneous apoptosis and retards colon tumor development in $Apc^{min716/+}$ mice, which highlights the critical role of Sam68-dependent NF-κB transactivation in the cellular responses to the intrinsic DNA damage that occurs frequently in the rapidly-dividing/proliferating cancer cells. We show here that $Khdrbs1^{-/-}$ mice suffer more severe damage in the colon and succumb rapidly from acute radiotoxicity than their $Khdrbs1^{+/-}$ controls post the extrinsic DNA damage challenge by WBIR. These results, extending additional support to the reported key role of NF-κB in providing radioprotection to the colon epithelium (*Egan et al., 2004*), highlight the pathophysiological relevance of the Sam68-dependent NF-κB activation in colonic cell survival and recovery from extrinsic/environmental DNA damage.

Elevation in Sam68 protein levels has been proposed as a prognostic marker in multiple cancers (*Chen et al., 2012*; *Liao et al., 2013*; *Song et al., 2010*; *Zhang et al., 2009*), although the exact function of Sam68 in these cancers remains obscure. We recently revealed that Sam68 plays a crucial role in controlling DNA damage-induced PARP1 activation and PAR production; hence Sam68 deficiency dramatically dampens the PAR-dependent NF-κB signaling and DNA repair pathways initiated by DNA damage (*Fu et al., 2016*; *Sun et al., 2016*). As a key early signaling regulator that converges at the proxy of the DNA damage-triggered signaling cascade in the nucleus, Sam68 could provide a novel target for cancer therapeutics. In support of this notion, manipulation of Sam68 sensitizes colon cancer to DNA damage-triggered apoptosis in human colon cancer cell lines and retards colon tumor burden in $Apc^{min716/+}$ mice (*Fu et al., 2016*). Besides its crucial role in cancer cells to overcome the frequently-occurred intrinsic DNA damage, our results here demonstrate that Sam68-dependent NF-κB transactivation is pivotal for normal cells in the colon epithelium by executing an important physiological function to prevent the radiodamage to the colon caused by extrinsic/environmental γ-irradiation. The levels of Sam68 proteins in both normal and cancerous colon tissues could be a potential biomarker to facilitate the optimization of the administered dose of γ-irradiation, when employed as a single therapy or combined with other means for cancer treatment, in order to achieve superior outcomes via an elegant balance between the antitumor effects to tumor tissue and the acute side-effects to normal tissue caused by γ-irradiation.

## Materials and methods

### Mice and ethics statement

All animal experiments were performed according to protocol number MO16-H285, approved by the Johns Hopkins University's Animal Care and Use Committee and in direct accordance with the NIH guidelines for housing and care of laboratory animals. $Khdrbs1^{-/-}$ mice and their gender-matched littermate $Khdrbs1^{+/-}$ mice were produced using heterozygous breeding pairs, as previously described (*Fu et al., 2016*). Mice were maintained in a specific pathogen-free facility and fed autoclaved food and water *ad libitum*.

### Whole-body γ-irradiation

Whole-body γ-irradiation (WBIR) in mice was performed as previously described (*Sun et al., 2016*). The γ-irradiated mice were sacrificed at indicated time points post WBIR for the indicated analyses, and the mortality and survival of mice were also monitored post γ-irradiation.

### Antibodies and reagents

Antibodies used were: Sam68 (RRID: AB_631869) and p65 (RRID: AB_632037) from Santa Cruz Biotechnology (Dallas, TX); $\beta$-actin (RRID: AB_476744) from Sigma-Aldrich (St. Louis, MO); PAR (RRID: AB_2572318) from Trevigen (Gaithersburg, MD); PARP1 (RRID: AB_2160739), phospho-p65 (RRID: AB_330570), Bcl-2 (RRID: AB_1903907), and cleaved Caspase-3 (RRID: AB_2341188) from Cell Signaling Technology (Danvers, MA); Bcl-XL (RRID: AB_1949733) from GeneTex (Irvine, CA). 4',6-diamidino-2-phenylindole (DAPI) was obtained from Sigma-Aldrich.

## Immunofluorescence staining

Immunofluorescence staining on colon tissue sections was performed as we did previous (*Fu et al., 2016*). Briefly, after euthanizing mice, the entire colons were excised under aseptic conditions and frozen in optimal cutting temperature (O.C.T.) media (Tissue-Tek, Elkhart, IN) or embedded in paraffin (Sigma-Aldrich). Tissue sections (5-micron) were cut, collected on coated slides, fixed in paraformaldehyde, washed with PBS, and blocked with appropriate sera in PBS. After incubating with appropriate antibodies, sections were washed and incubated with fluorescence dye-conjugated second antibodies and 1 μg/ml of DAPI (Sigma-Aldrich). Stained sections were washed and mounted under a coverslip using Fluoro-gel with Tris Buffer (Electron Microscopy Sciences, Hatfield, PA) and examined using an Axio Observer fluorescence microscope (Zeiss, Oberkochen, Germany).

## Isolation of primary colonic epithelial cells

Colonic epithelial cells (CECs) were isolated from mice as previously described (*Fu et al., 2016*; *Hodgson et al., 2015*).

## Subcellular fractionation

Subcellular fractionation was performed by differential centrifugation as previously described (*Wan et al., 2007*; *Wier et al., 2012*).

## Immunoblot

Immunoblot assays were conducted as previously described (*Fu et al., 2013*; *Hodgson et al., 2015*). In brief, cells were harvested and lysed on ice by 0.4 ml of lysis buffer (50 mM Tris-HCl [pH 8.0], 150 mM NaCl, 1% NP-40 and 0.5% sodium deoxycholate, 1 × complete protease inhibitor cocktail [Roche Applied Science, Indianapolis, IN]) for 30 min. The lysates were centrifuged at 10,000 × $g$ at 4°C for 10 min. The protein-normalized lysates were separated by SDS-PAGE under reduced and denaturing conditions. The resolved protein bands were transferred onto nitrocellulose membranes and probed by the Super Signaling system (Thermo Scientific) according to the manufacturer's instructions, and imaged using a FluorChem E System (Protein Simple, Santa Clara, CA).

## mRNA in situ hybridization

Digoxigenin (DIG)-labeled probes were employed to visualize *Bcl2l1* mRNA encoding Bcl-XL in colon tissues, as previously described (*Hobbs et al., 2015*). Briefly, *Bcl2l1* gene specific sequence was first ligated to the pCRII-TOPO Vector (Life Technologies). The antisense and sense complementary RNA probes specific for *Bcl2l1* mRNA were transcribed using a Lig'n Scribe Kit (Life Technologies), and then labeled with DIG using a DIG RNA labeling kits (Roche Applied Science) according to the manufacturer's instructions. The mRNA in situ hybridization on frozen colon tissue sections was performed using adapted methods from *Gu and Coulombe (2007)*. Briefly, colon tissues were postfixed in 4% paraformaldehyde/PBS for 20 min, followed by proteinase K digestion at 37°C for 6 min and re-fixed in 4% paraformaldehyde/PBS, then acetylated by 0.25% acetic anhydride in 0.1 M triethanolamine (10 min). Hybridization solution containing 3 μg of each denatured DIG-labeled probe was mixed with the samples for overnight incubation at 65°C. The next day, slides were rinsed and incubated in the HSW solution (50% formamide, 0.5 × standard sodium citrate, 0.1% Tween-20) for 30 min at 65°C. The slides were then washed in the HSW solution (2 × 20 min) at 65°C, 2 × standard sodium citrate, 0.1 × standard sodium citrate at 37°, respectively. The slides were switched to blocking solution (10% normal goat serum [NGS] in PBST) for 1 hr, followed by an incubation in alkaline phosphatase (AP)-conjugated sheep anti-DIG-antibody (Roche Applied Science), 1:2000 diluted in PBST/1% NGS overnight at 4°C in the dark. To visualize the mRNA in situ hybridization signal, tissues were washed with PBST (3 × 2 hr) and NTMT (0.1 M NaCl, 0.1 M Tris-HCl [pH7.9], 50 mM MgCl₂, 0.1% Tween-20) for 10 min, and incubated in BM purple AP-substrate (Roche Applied Science) containing 0.5 mg/ml levamizole overnight, then stopped the reaction by washing in PBS. Hybridized tissues are mounted in crystal/mount media in preparation for microscopy.

## TUNEL assays

Terminal deoxynucleotidyl transferase dUTP nick end labeling (TUNEL) in situ on colon tissue sections were carried out using a DNA fragmentation Image Kit (Roche Applied Science), according to the manufacturer's instructions.

## Histology

Histological analyses were carried out as we did previously (*Fu et al., 2016*). In brief, the excised entire colons were embedded in paraffin. Tissue sections (5-micron) were cut, deparaffinised, rehydrated, and stained with hematoxylin and Eosin (H and E) staining and stained sections were washed and mounted under a coverslip and examined under light microscopy (Zeiss). The crypt survival assays (*Lai and Egan, 2013*) were employed to evaluate the radio-sensitivity of the colon post whole-body γ-irradiation in mice.

## Statistical analyses

All statistical analysis was performed using GraphPad Prism version 6.0 (GraphPad Software, La Jolla, CA). Standard errors of means (s.e.m.) were plotted in graphs. Significant differences were considered: ns, non-significant difference; * at $p<0.05$; ** at $p<0.01$; *** at $p<0.001$; **** at $p<0.0001$ by unpaired Student's $t$-test.

## Acknowledgements

We thank Xin Guo for help with histological analyses.

# Additional information

### Funding

| Funder | Grant reference number | Author |
| --- | --- | --- |
| National Institute of General Medical Sciences | R01GM111682 | Fengyi Wan |
| American Cancer Society | RSG-13-052-01-MPC | Fengyi Wan |
| National Cancer Institute | T32CA009110 | Eric M Wier<br>Ryan P Hobbs |

The funders had no role in study design, data collection and interpretation, or the decision to submit the work for publication.

### Author contributions

KF, Designed and conducted most experiments; Acquisition of data; Analysis and interpretation of data; XS, EMW, AH, Helped with some experiments; Acquisition of data; Analysis and interpretation of data; RPH, Helped with some experiments; Analysis and interpretation of data; Contributed unpublished essential data or reagents; FW, Conceptualized the study, designed the experiments, and wrote the manuscript with input from all authors; Analysis and interpretation of data

### Author ORCIDs

Xin Sun, http://orcid.org/0000-0003-2424-8011
Fengyi Wan, http://orcid.org/0000-0001-9216-9767

### Ethics

Animal experimentation: All animal experiments were performed according to protocol number MO16-H285, approved by the Johns Hopkins University's Animal Care and Use Committee and in direct accordance with the NIH guidelines for housing and care of laboratory animals. Khdrbs1-/- mice and their gender-matched littermate Khdrbs1+/- mice were produced using heterozygous breeding pairs. Mice were maintained in a specific pathogen-free facility and fed autoclaved food and water ad libitum.

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
