## [Decision Letter]

Thank you for submitting your article "Sam68/KHDRBS1-dependent NF-κB activation confers radioprotection to the colon epithelium in γ-irradiated mice" for consideration by *eLife*. Your article has been reviewed by two peer reviewers, and the evaluation has been overseen by Tony Hunter as the Senior Editor and Reviewing Editor. The following individual involved in review of your submission has agreed to reveal his identity: David Levens (Reviewer #1).

The reviewers have discussed the reviews with one another and the Reviewing Editor has drafted this decision to help you prepare a revised submission.

Summary:

This is a nice follow up of the authors' recent *eLife* paper in which they reported a role for the Sam68 RNA binding protein in NF-κB activation in response to DNA damage in cultured cells. The new data show quite convincingly that in response to whole body irradiation (WBIR) Sam68-/- mice exhibited reduced induction of NF-κB target genes, such as BCl^-^XL, decreased PARylation of nuclear proteins, increased Casp3 cleavage and apoptosis in colonic epithelial cells compared to Sam68+/- mice. They also showed that the absence of Sam68 resulted in a significant increase in the percentage of mice dying following WIBR, and concluded that Sam68 is required for protection against colonic damage in response to irradiation.

This is an important extension of their previous in vitro results, and the data show convincingly that Sam68 has a clear radioprotective role in the colon epithelium in vivo by controlling NF-κB activity.

Please address the following minor points in a revised version:

1) In the Introduction, the authors call Sam68 an RNA-binding protein. It would be more correct and proper to call it a single-strand nucleic acid binding protein. This distinction is important as the authors have not yet implicated RNA in the pathway of DNA damage, and so it is entirely possible that the pathway is initiated or involves binding with ssDNA exposed during the damage response at an early stage.

2) The writing can be improved. It is in places unclear – for example in the Results section, the authors write "…*Khdrbs^+/-^* and *Khdrbs^-/-^*mice exhibited similar results (Figure 1)", but this is not at all what they meant. The two genotypes yield different results; what the authors meant is that Figure 1 report similar conclusions. The paper needs to be carefully re-read for precision and clarity.

3) The order in which the WT and KO samples are portrayed in the figures is switched in Figure 3 and Figure 4 when compared to Figure 1 and 2. It is suggested that the sample order be rearranged so that it is uniform across all figures.

---

## [Author Response]

*[…] Please address the following minor points in a revised version:*

*1) In the Introduction, the authors call Sam68 an RNA-binding protein. It would be more correct and proper to call it a single-strand nucleic acid binding protein. This distinction is important as the authors have not yet implicated RNA in the pathway of DNA damage, and so it is entirely possible that the pathway is initiated or involves binding with ssDNA exposed during the damage response at an early stage.*

We appreciate the reviewers’ suggestion and agree on that “single-strand nucleic acid binding protein” would be more correct and proper for Sam68. We have made the revision in the Introduction section.

2) The writing can be improved. It is in places unclear – for example in the Results section, the authors write "…Khdrbs^+/-^ and Khdrbs^-/-^ mice exhibited similar results (Figure 1)", but this is not at all what they meant. The two genotypes yield different results; what the authors meant is that Figure 1 report similar conclusions. The paper needs to be carefully re-read for precision and clarity.

We appreciate the reviewers for pointing out that our description of the results in Figure 1 was unclear. As suggested, we have polished the English and double-checked the precision and clarity of the writing throughout the manuscript.

*3) The order in which the WT and KO samples are portrayed in the figures is switched in Figure 3 and Figure 4 when compared to Figure 1 and 2. It is suggested that the sample order be rearranged so that it is uniform across all figures.*

We thank the reviewers’ suggestion to arrange the sample order in Figure 3 and Figure 4, and now the order in which the WT and KO samples is uniformed across all figures.